# Duplicate Detection of Spike Events: A Relevant Problem in Human Single-Unit Recordings

**DOI:** 10.3390/brainsci11060761

**Published:** 2021-06-08

**Authors:** Gert Dehnen, Marcel S. Kehl, Alana Darcher, Tamara T. Müller, Jakob H. Macke, Valeri Borger, Rainer Surges, Florian Mormann

**Affiliations:** 1Department of Epileptology, University of Bonn Medical Center, Venusberg-Campus 1, 53127 Bonn, Germany; gert.dehnen@ukbonn.de (G.D.); marcel.kehl@ukbonn.de (M.S.K.); alana.darcher@ukbonn.de (A.D.); rainer.surges@ukbonn.de (R.S.); 2Computational Neuroengineering, Department of Electrical and Computerengineering, TU Munich, 80333 Munich, Germany; tamara.mueller@tum.de (T.T.M.); jakob.macke@uni-tuebingen.de (J.H.M.); 3Institute for Artificial Intelligence and Informatics in Medicine, TU Munich, 80333 Munich, Germany; 4Machine Learning in Science, University of Tübingen, 72076 Tübingen, Germany; 5Department of Neurosurgery, University of Bonn Medical Center, Venusberg-Campus 1, 53127 Bonn, Germany; valeri.borger@ukbonn.de

**Keywords:** human single-unit recordings, artifact removal, spike sorting

## Abstract

Single-unit recordings in the brain of behaving human subjects provide a unique opportunity to advance our understanding of neural mechanisms of cognition. These recordings are exclusively performed in medical centers during diagnostic or therapeutic procedures. The presence of medical instruments along with other aspects of the hospital environment limit the control of electrical noise compared to animal laboratory environments. Here, we highlight the problem of an increased occurrence of simultaneous spike events on different recording channels in human single-unit recordings. Most of these simultaneous events were detected in clusters previously labeled as artifacts and showed similar waveforms. These events may result from common external noise sources or from different micro-electrodes recording activity from the same neuron. To address the problem of duplicate recorded events, we introduce an open-source algorithm to identify these artificial spike events based on their synchronicity and waveform similarity. Applying our method to a comprehensive dataset of human single-unit recordings, we demonstrate that our algorithm can substantially increase the data quality of these recordings. Given our findings, we argue that future studies of single-unit activity recorded under noisy conditions should employ algorithms of this kind to improve data quality.

## 1. Introduction

The opportunity to record from single neurons in the brain of behaving human subjects increasingly contributes to the advances of cognitive and systems neuroscience. These recordings allow researchers to investigate complex brain functions, such as perception [1,2,3,4,5,6,7,8,9], memory [10,11,12,13,14], emotion [15,16], or decision making [17,18].

In these studies, humans are implanted with intracranial electrodes solely for medical purposes, such as the identification of the seizure onset zone in patients with pharmacologically intractable epilepsy [19], the treatment of movement disorders [20,21], or the management of treatment-resistant depression [22]. In some medical centers, it is currently possible to record the activity of single neurons from these patients while performing cognitive experiments. As the opportunity to perform such experiments is exceedingly rare, it is imperative that researchers optimize the quality of the recorded data. In a clinical setting, there are many different external sources of noise, such as medical instruments close to the patient, and only limited possibilities to control the setting [23]. The signal quality can be increased during the recordings by eliminating local noise sources, as well as after the recording by detecting artifacts using advanced spike-sorting algorithms. In animal studies, the development of polytrodes has increased the signal-to-noise ratio and thereby the reliability of single-unit recordings [24,25,26]. Since polytrodes have not yet been accredited for use in humans, most medical centers currently use microwire bundles, which have a considerably lower signal-to-noise ratio. To take full advantage of these datasets, it is essential that researchers identify as many artifacts and sources of noise as possible in microwire recordings.

Despite the limitations mentioned above, various advances in electrophysiological technology have led to a rapid increase in the total number of neurons recorded in a given experiment. To deal with this increasing amount of data, automated spike-sorting algorithms have become crucial to efficiently extract and cluster the neuronal spike events. In the past two decades, numerous spike-sorting algorithms have been developed (e.g., [26,27,28,29,30,31,32,33,34,35,36,37,38,39,40,41,42,43,44]) with explicit focus on the reliability of extracted units and the quality of their separation. Some automated sorting algorithms try to increase data quality by using different criteria to detect artificial events (e.g., [27,28,29,30,31,32,33,34,35]). However, even algorithms adjusted to human single-unit recordings include only elementary methods for the identification and removal of artificial events [27,28,29,41]. An important and commonly overlooked artifact is the recording of spike events simultaneously in multiple channels. So far, no method has been developed to address this issue in human microwire recordings. We therefore propose a new method for detecting events occurring simultaneously in multiple channels with similar features. We present our method as an open-source and freely available MATLAB module that allows researchers to further improve the results of spike sorting from existing cluster algorithms (Combinato Spike Sorting [29]; Wave_clus [27,41]) by removing duplicate spike events.

### 1.1. Motivation

Recording single units in humans entails a clinical setting and thus a noisy environment from which it is impossible to eliminate all sources of artifacts. In order to optimize the data quality and unit yield, it is crucial to distinguish neural spike events from artificial spike events. Below we discuss prominent sources of noise and the characteristics of these artificial events.

Often, technical aspects of the recording set-up itself can lead to artificial spike events, such as electrical interference, head and cable movements, or broken wires [23]. All of these technical issues originate from non-neural sources and can produce artifacts that may be recorded on several channels simultaneously.

Additionally, duplicate spike events can also originate from physiological sources. For arrays [45], or bundles of microwires, individual micro-electrodes can end up rather close to one another. Since their individual recording volumes can overlap, the same unit can be recorded on different channels [46,47]. Another potential physiological cause of duplicate spike events is a biphasic shape of the signal that is extracted from the same channel exceeding the positive and negative extraction thresholds [31,33,47]. Moreover, a spike event may be recorded on a reference wire and thus appear inverted on multiple channels referenced against this wire [23]. Complicating the matter, physiological coincident spikes may also occur naturally as a reflection of temporal coding in local neuronal networks [48]. Therefore, it is crucial to distinguish between artificial and physiological or pathological coincident spike events.

#### 1.1.1. Artifact Detection Methods

In extracellular recordings of action potentials (‘spikes’), various approaches have been used to identify artificial spike events. Most of these approaches can be classified into the following categories:1.*Artifact detection based on spike shape*

As we have a fairly good understanding of the spike shapes of physiological action potentials [49], the shape of a spike event can provide information on its origin. For example, artificial spike events produced by electrical interference often exhibit a sinusoidal shape [29]. Furthermore, artificial events exhibiting an unphysiologically high amplitude [29,31] can be easily detected.
2.*Artifact detection based on spike timing*

The time course of spike events in a potential unit appearing as a cluster in a spike-sorting algorithm can provide further indications of its origin. For instance, if this cluster contains a high proportion of inter-spike intervals below the physiological refractory period, then these spikes cannot originate from one single unit but instead must be contributions by other units’ activity or by artifacts [29]. A different approach is to analyze simultaneous spike events across different channels. Several spike-sorting algorithms identify redundant clusters (e.g., [31,33]) and remove noise outliers in the frequency domain depending on the spatial resolution of electrodes in an array [33].
3.*Manual artifact detection based on spike shape and timing*

Experienced operators can combine spike-shape and spike-timing information to label clusters as artifacts. However, manual evaluation of a large number of clusters is rather time-consuming, especially for a large number of channels, and typically limited to a given recording channel. Effects and interactions occurring simultaneously in different channels are therefore typically disregarded. Recently, there have also been approaches using deep-learning classifiers to automate this process [34,40,43,44,50].

If the spatial configuration of electrodes in polytrodes or arrays is known, then this information can be used to differentiate more reliably between local neural spike events and artifacts [26,30,31,33,35,39,40,42]. Due to the flexible nature of microwire bundles, it is usually not possible to infer their precise spatial configuration.

#### 1.1.2. Characteristics of Coincident Spike Events in Human Single-Unit Recordings

The previous section has demonstrated the need for effective strategies to detect duplicate spike events. To develop our methods, we first looked at the characteristics of simultaneous spike events with respect to their temporal synchrony and shape.

Close examination of our recorded data revealed that simultaneous spike events in different channels often exhibit very similar event shapes (examples in Figure 1A and Figure 2B). Interestingly, this observation was made for simultaneous spike events within the same wire bundle (spike events marked in blue in Figure 1A) as well as spike events occurring in different wire bundles (spike events marked in red in Figure 1A). Furthermore, we found that spike events of the real data occurring within a small time bin show a significantly higher proportion of similar event shapes than spike events from different hemispheres at different time bins (*p* = 3.3 × 10^−7^, Wilcoxon signed-rank test, ∆*t* = 50 µs, median of real population against surrogate population per session, see Section 2.3.2.). This finding demonstrates that simultaneous spike events exhibit more similar shapes.

Regarding the temporal synchrony, Figure 1B illustrates an example of binned event counts in a 15 s data segment pooled across all 80 recording channels. A substantial fraction of the 0.5 ms bins contain two or more spikes. A large proportion of these bins exceed the red line, indicating the mean + 5 standard deviations (σ) of the spike-count-distribution of the original data. It is worth noting that these bins show a temporal clustering (see, e.g., Figure 1B, original data around *t* = 7 s) that likely corresponds to time intervals of poor data quality (e.g., movement artifacts). In order to estimate the rate of coincident spike events, we generated a time-shifted surrogate based on the original data by circularly shifting all extracted spike events, independently for each cluster, by some random offset. This procedure eliminates all effects of simultaneity in the dataset [51]. In this example, none of the bin counts in the time-shifted surrogate exceed the 5σ threshold of the original data, demonstrating that simultaneous spike events occur above chance.

In order to investigate whether this effect generalizes across recording sessions, we next calculated the frequency of simultaneous spike events across all recorded data used in this study. For this purpose, we counted the number of spike events in each bin across all recordings. Figure 1C shows the abundance of bins filled with different numbers of spike events. Applying the same procedure to 1000 time-shifted surrogate data yielded a distribution for the abundance of bins filled with different numbers of spike events without temporal synchronization of the recorded clusters. For this surrogate dataset, the proportion of these bins decreased rapidly compared to the original dataset. This reveals that our recorded data contain a significantly increased number of simultaneous spike events (see Figure 1C).

In order to better understand from where the spike events in these bins originate, we manually classified clusters based on our spike-sorting algorithm [29] into single units (SU), multi-units (MU), and artifacts (Art) (for details, see Section 2.1.). Figure 1D illustrates the mean proportion of cluster types contributing to the spike events in bins with two or more spikes, averaged across recordings. Time bins with high event counts contain more artifact spikes than bins that are filled with only two simultaneous events. This indicates that spike events occurring simultaneously in numerous channels most likely belong to artifact clusters.

Using these characteristics of spike events in real data, in the following section, we present our algorithm to detect duplicate artificial events.

## 2. Materials and Methods

### 2.1. Data and Materials

In order to develop and optimize our algorithm, we used a dataset of 51 recording sessions from 13 patients with pharmacological intractable epilepsy (for details see Table 1). For diagnostic reasons, patients were implanted bilaterally in the medial temporal lobe (MTL) with 9–12 Behnke-Fried depth electrodes (AD-TECH Medical Instrument Corp., Racine, WI, USA). The exact locations of depth electrodes were exclusively defined for clinical diagnostics. Each electrode contained eight high-impedance and one low-impedance platinum-iridium wires spreading out at the end of its tip. Using these eight high-impedance micro-contacts, we were able to record action potentials from single units. The ninth low-impedance wire was used as a recording reference. The data were collected with an ATLAS recording system (Neuralynx Inc., Bozeman, MT, USA). All data were referentially recorded, filtered at a frequency range of 0.1–9000 Hz, and sampled at 32,768 Hz. We analyzed 51 recording sessions lasting about 25 min each (mean: 25.6 min, *SD*: 3.6 min), that were used to screen for visually responsive units [6,8]. After data collection, spike events were automatically extracted, sorted, and manually evaluated using the Combinato software package [29] (see Appendix A, Table A1). Spike events with positive and negative deflections in Combinato were sorted separately using the default parameters from [29]. Each extracted event shape was sampled by 64 data points spanning a time window of 2 ms. Using the Combinato software package, the sorted clusters were automatically labeled as artifacts or multi-units. During manual evaluation the automated sorting results were checked and optimized, and units were classified as single units, multi-units, or artifacts based on their firing characteristics and spike shapes (see e.g., [46,47]). 

Units showing characteristic peaks in their inter-spike interval (*ISI*) histograms stemming from electrical sources (e.g., 50 Hz line noise resulting in peaks at multiples of 20 ms), or having an unphysiological spike-event shape, that were not automatically labeled as artifacts, were merged together with all other artifacts into one artifact cluster. To label a cluster as a single unit, several conditions had to be fulfilled: a physiological waveform in the density plot with a well-defined shape and a steep increase; an asymmetrical spike-event shape with respect to the maximum of the mean cluster shape, and an *ISI* < 3 ms for less than 5% of all spike events in a cluster. Units that were not labeled as artifacts and did not meet all of the above criteria were labeled as multi-units.

As we are specifically interested in artifacts during human single-unit recordings in a clinical setting, in the following regard, these data serve as a gold-standard since there is no ground truth data available for these types of recordings. 

### 2.2. Structure of the Duplicate Event Removal Algorithm

To identify spike events that are spuriously recorded or detected multiple times, we implemented the duplicate event removal algorithm (DER algorithm) consisting of three parts (see Figure 2):**Part I**. Detect simultaneous spike events between different bundles.**Part II**. Identify duplicate detected biphasic spike events on the same channel and simultaneous spike events on different channels within the same bundle.**Part III**. Detect duplicate spike events based on unphysiologically high zero-lag cross-correlation between clusters.

The open-source code of the DER algorithm and further instructions are accessible at GitHub (https://github.com/Geaht/DER, accessed on 20 May 2021).

### 2.3. Part I—Detection of Artifacts across Bundles

Sources of noise originating from the environment and the clinical setup are often recorded on several bundles simultaneously. As these artifacts can lead to spike events that look very similar to action potentials of neurons (see Figure 1A and Figure 2C), they can be extracted by automatic spike-sorting algorithms but may not be labeled as artifacts. Part I of the DER algorithm identifies these artifacts by detecting spike events of similar shape recorded in different bundles at the same time.

#### 2.3.1. Detection of Simultaneous Artifacts

If more than two spike events (*no*_sim_) appeared within a time window of Δ*t*_max_ = 50 µs (see Section 2.3.2) in two or more different bundles, we compared the shape of each pair of spike events. We extracted the features of each spike-event shape with a five-level discrete wavelet decomposition (Haar wavelets). Next, we reduced the dimensionality of the feature space to the 10 dimensions in which the distribution of the wavelet coefficients differed most strongly from a normal distribution (quantified with a Kolmogorov–Smirnov test statistic). This feature extraction of spike-event shapes is motivated by the feature extraction used by the spike-sorting algorithms Wave_clus and Combinato [27,29]. For each spike-event pair, the Euclidean distance of their selected wavelet coefficients was calculated as a measure of shape similarity. If the median Euclidean distance of all combinations of events within this time window was below the threshold of *d*_thr_ = 14.6 (see Section 2.3.2 and Table 4), all spike events in this time bin were labeled as artifacts, since only these spike events are likely to fulfill both criteria.

#### 2.3.2. Definition of Thresholds of the Euclidean Distance and the Time Window

In Part I and Part II of the algorithm, two thresholds are needed to define spike events appearing simultaneously with a similar shape: a maximum difference in occurrence time (as a measure of simultaneity) and a maximum Euclidean distance (representing the similarity of two event shapes).

The best proxy for actual duplicate artifacts in our data are spike events occurring simultaneously in different bundles that have been manually labeled as artifacts during the clustering process. To define a threshold for the similarity of coincident artificial spike events in different bundles, we compared two populations of previously labeled artifacts.

From these artifacts we randomly chose 10,000 pairs of spike events per recording session from different hemispheres that did not occur within a time window of ∆*t* (surrogate population) and 10,000 spike-event pairs from different bundles that did occur within this time window (real population). The corresponding two distributions of Euclidean distances (see Figure 3A) were used to calculate the ROC (receiver operating characteristics) curve shown in Figure 3B. Going in steps of 0.1 from zero to the highest Euclidean distance of both populations, each point in the ROC curve was calculated by counting the number of true and false positives as well as true and false negatives. The point on the ROC curve with minimal distance from the point (0,1) was chosen as the operating point (see [52]), representing the best threshold (*d*_thr_) to separate the two distributions.

To define the threshold for the maximum time lag of spike events considered as simultaneous, we used these two populations of artifacts and calculated for different time windows (32 µs–1 ms) the operating point in the ROC curve and the area under the ROC curve (AUC) (see Figure A1). The best separation of these two populations was achieved by maximizing the AUC leading to small time windows, especially Δ*t*_max_ = 32 µs and Δ*t*_max_ = 50 µs. As there was virtually no difference in AUC for these two thresholds, we chose Δ*t*_max_ = 50 µs as detection threshold in time, yielding a threshold of the Euclidean distance of *d*_thr_ = 14.6 (Table 4).

### 2.4. Part II—Duplicate Events in the Same Bundle

Part II of our detection algorithm focuses on neuronal spike events that were extracted multiple times on the same channel or recorded simultaneously on different wires in the same bundle.

#### 2.4.1. Same Channel

The positive and negative amplitude of biphasic spikes can cross the positive and negative extraction thresholds leading to the same spike event being extracted twice with opposite polarity. Previous studies [53,54] have distinguished putative interneurons from principal cells based on spike duration, using a classification threshold of 650 µs from a spike-shape’s peak to its trough (see all parameters in Table 4). As interneurons tend to have a biphasic spike shape, we used this time window to detect duplicate spike events of biphasic shapes. If two spike events on the same channel are detected within that time window and have opposite signs in their amplitudes, they are labeled as duplicate spikes. To decide which of the two spike events to retain and which to label as artificial, we use the following criteria based on the existing unit classification:If one of the two duplicate events was labeled as an artifact by an automated clustering algorithm or by manual reclustering, this event is labeled as artificial (Table 2, Case 1).If the two events have different unit labels (i.e., one is a single- and the other a multi-unit), we keep the spike event from the single-unit and mark the other one as an artifact (Table 2, Case 3).If both spike events are of the same unit class (both single- or multi-unit), we calculate the signal-to-noise ratio (SNR, peak amplitude/spike extraction threshold) of each event and keep the event corresponding to the higher value (Table 2, Case 4). For further details about the thresholds of spike extraction, see [27,29].

#### 2.4.2. Same Bundle

In the next step, we investigated physiological and non-physiological duplicate recorded spike events within the same bundle.

To detect spike events recorded simultaneously in different channels of the same bundle, we looked again for spike events that appeared within a short time interval. For any pair of events fulfilling this criterion, we compared their shape (the Euclidean distance for each pair) as described in Section 2.3.1. To compare event shapes within the same bundle, we took into account that these spike events might be caused by neural action potentials recorded on more than one microwire. One of the criteria to label a unit as single- or multi-unit is its cluster shape. If the cluster shape is in accordance with a physiologically expected signal [49], this indicates a neuronal origin. Consequently, single- and multi-units have a smaller variety of cluster shapes than artifacts whose origin may vary (see Section 1.1). Therefore, we calculated the thresholds for the detection of duplicate spike events in the same bundle according to Section 2.3.2, but using two populations of SU and MU (SU/MU within the same bundle compared to SU/MU from different hemispheres). This led to the same maximum time difference as in Part I (Δ*t*_max_ = 50 µs) for two spike events to be considered as duplicate detections, and a resulting threshold for their Euclidean distance of *d*_thr_ = 8.4 (see Figure 3C,D, Figure A1 and Table 4). If the compared spike events fulfilled these criteria, we decided which spike event to retain using similar criteria as in Section 2.4.1 (see Table 2, Case 2–4).

### 2.5. Part III—Cross-Correlations

A standard method for assessing relationships in the firing patterns of two recorded units is provided by analyzing their cross-correlation (e.g., [55]). In our dataset we identified multiple cross-correlations exhibiting a prominent increase in the central time bin of the cross-correlograms (see e.g., Figure 2E). Besides physiological synchronization, these cross-correlations might originate from simultaneous artifact events on different channels or from units that are recorded with more than one microwire.

#### 2.5.1. Calculation of all Cross-Correlations

In order to identify potential spurious cross-correlations across all possible pairs of recorded units, we calculated for each recording session a cross-correlation matrix *C* (*N*_cluster_ × *N*_cluster_ × *N*_time-bins_). The cross-correlogram of two clusters *i* and *j* can be obtained from *C* as *C*(*i,j,:*). To achieve nearly linear computational scaling with increasing spike-event counts, our algorithm loops only once through a list of all time-ordered spike times of a recording session. Each spike is checked for subsequent spikes within a maximal time lag of the cross-correlation *t*_max_ (e.g., 20.25 ms for *t*_bin_ = 0.5 ms; *N*_bins_ = 81; *t*_max_ = ½ *t*_bin_ ∙ *N*_bins_). For each spike event detected with a time delay of Δ*t* < *t*_max_, the count in the corresponding time bin of *C* is increased by 1. Since cross-correlations are skew symmetric (*C*_ij_ = −*C*_ji_), it is sufficient to consider only subsequent spike events and complete the cross-correlation matrix afterwards. The magnitude of the spike-event counts within the central time-bin (e.g., from −250 µs to +250 µs) can be assessed by calculating a z-score for the central bin based on the mean and standard deviation of spike-event counts in all other time bins of the cross-correlogram. This method yields one z-score for each combination of recorded clusters (see Figure 3G).

#### 2.5.2. Detection of Suspicious Cross-Correlations

If the spiking of two recorded neurons were independent, the cross-correlogram would be expected to be flat without an asymmetry or a central peak. However, there are several physiological and technical reasons that could cause increased simultaneous firing of two units: A possible physiological reason for increased simultaneous firing are two neurons receiving a direct synaptic input from a third neuron (e.g., [55]). An asymmetry in the cross-correlogram can be caused by a direct or indirect synaptic connection between the two neurons [56]. Moreover, sensory inputs may also cause increased simultaneous firing of different neurons, even without direct synaptic connections [57]. Beside physiological reasons, there are several technical reasons that can also lead to an increased number of simultaneous spike events (see Section 1.1). This can result in a high z-score of the central bin of the cross-correlogram.

In order to identify spike events originating from one of the sources described above, we propose the following procedure: First, identify cluster pairs that exceed a given z-threshold for the spike-event count in the central bin (*z*_central_ > 5, see Section 2.5.3). Depending on the cluster pair combination, spike events within the central bin of the cross-correlogram are labeled according to the scheme in Table 3. Spike events labeled this way might be considered for later deletion as they are likely caused by artifacts (Cases 1 & 2) or represent duplicate recordings of the same neuronal spike events (Cases 3 & 4).

#### 2.5.3. Threshold for the Central Bin of the Cross-Correlogram

For the method described above, a threshold *z*_thr_ must be determined. Coincident spike events in two given clusters which exceed this threshold are considered artificial or duplicate. Such a threshold can be derived from our recorded data by using the distributions of central z-values across different cluster pairs. Artifact clusters within the same wire bundle were recorded from the same anatomical region and shared the same processing electronics (wires, connectors, etc.). Therefore, they are expected to contain a high proportion of coincident spike events. In contrast, single units recorded from different hemispheres should not contain spurious duplicate spike events. In our dataset, we indeed found that the central z-values of SU pairs originating from different hemispheres were significantly smaller than the z-values from pairs of artifact clusters within the same region (*p* = 8.9 × 10^−16^, Wilcoxon signed-rank test of the medians across sessions, *N*_Sessions_ = 51).

We analyzed how well these two distributions (Figure 3E) can be separated by a single threshold *z*_thr_. All counts below this threshold may be considered to belong to the SU pairs, while all the others are assigned to the artifact pairs. This procedure allowed us to calculate the number of true and false positive as well as true and false-negative counts for different threshold values *z*_thr_. The resulting sensitivity and specificity values are plotted as a ROC curve (Figure 3F). The minimal distance to the point (0,1) is given by the operating point at *z* = 3 (as used in Figure 3B,D, see e.g., [52]). This indicates that cluster pairs with a central z-value above 3 in the cross-correlogram resemble pairs of artifact clusters within the same wire bundle rather than independent units. To avoid false positive detections of artifacts (i.e., spuriously discarding neuronal spikes as artifacts), we chose a more conservative threshold of *z*_thr_ = 5 for our algorithm (Table 4). Spike events occurring within this central bin of the cross-correlogram are labeled following the description in Table 3.

For further validation, we analyzed the type of events detected in Part III of our algorithm by identifying from which type of cluster (SU/MU/artifact—as determined by the spike-sorting procedure) they originated. Figure 3I displays the percentage of artifacts, MU, and SU that were detected, averaged across all 51 recording sessions, using different thresholds for the central z-value. For instance, for *z* = 5, about 35% of all manually clustered artifact events are detected by Part III of our algorithm, while only 13% of the MU and just 9% of the SU events were labeled as duplicate spike events. Remarkably, the percentage of artifacts labeled by the algorithm strongly decreases for higher *z* thresholds while the percentage of labeled MU and SU decreases only marginally. These observations demonstrate that most of the detected spike events correspond to artificial events.

## 3. Results

In this section, we report the performance of our algorithm on real data. First, we present two examples of data that are contaminated by duplicate spike events and demonstrate the improvement achieved by the DER algorithm. We then illustrate the proportions of detected duplicate spike events for the different types of clusters as well as for the different parts of the DER algorithm.

### 3.1. Examples of Improved Data Quality

Figure 4A shows three raster plots of the same 30 s data segment for all clusters (including clusters labeled manually as artifacts) from two wire bundles in the left posterior hippocampus and entorhinal cortex. The left panel of this figure shows the original data before the DER algorithm. Note the simultaneous spike events on every wire, occurring approximately seven seconds after the beginning of the recording. These spike events are likely caused by external noise sources as they appear on different bundles in parallel. The middle panel illustrates spike events marked by the different parts of the DER algorithm. The suspicious spike events around the seventh second in the raster (and several others) are detected by the algorithm. To illustrate the resulting raster plot we show in the right panel all remaining spike events. A comparison of the raster plots before (left) and after (right) duplicate event removal indicates that the majority of suspicious synchronous spike events were removed, and the data quality was enhanced.

A second example of the improvement in data quality is shown in Figure 4B. We noticed on several occasions that units of similar shape from different microwires within the same wire bundle responded to the same stimuli [2]. Illustrating this phenomenon, Figure 4B (upper two panels; spike events are marked according to the different parts of the DER algorithm using the same colors as in Figure 4A) shows two single units recorded in the right posterior hippocampus (RPH2 and RPH4) within the same bundle, but on different microwires. The first unit (RPH2) shows a clear response to the image of the German comedian Otto Waalkes (stimulus 2). The second unit (RPH4) primarily responds to the German singer Helene Fischer (stimulus 1), but also increases its firing-rate in response to Otto Waalkes. Note that the spikes of the second unit occur at similar time points as the spikes of the first unit during presentation of the second stimulus. This overlap in the response behavior and the similarity of their cluster shapes (see density plots in Figure 4B) hint towards this being a duplicate recording of neuronal spike events.

After applying the DER algorithm to these data, most spike events during the response period in the second unit are detected as duplicate events and are therefore removed (see red framed raster in Figure 4B), while both primary responses remain unchanged. This example highlights the importance of detecting duplicate spike events when investigating the response behavior (e.g., selectivity) of concept cells. Furthermore, statistics in single-unit studies are commonly performed across the population of all recorded units, and single units are often analyzed independently (e.g., [1,2,3,4,5,6,7,8,9,10,11,12,15,16,17,18]). 

Our algorithm systematically reduces dependencies across recorded units which may originate from duplicate detected spike events.

### 3.2. Overall Performance of the DER Algorithm

To assess the overall performance of the DER algorithm, we applied it to the complete dataset of 51 recording sessions and analyzed how many spike events were detected by each part of the algorithm. We further separated the detected spike events based on the type of cluster to which they belonged (SU, MU, artifacts).

Altogether, our algorithm marked more than a fifth of all recorded spike events (22.05%) as duplicate and/or artificial (Figure 4C and Table 5). This proportion is rather high since we included all artifact clusters as well as spike detection with positive and negative deflection. In particular, spike events with negative deflections in Combinato were significantly more often detected than spike events with positive defections (50.44% vs. 16.13%, *p* = 8.9 × 10^−16^, Wilcoxon signed-rank test across recordings; *N* = 51).

Interestingly, the detection of biphasic spikes within the same channel (Part II—Same channel) deleted 6.46% of all spikes, which were almost exclusively (98.25%) found in artifact clusters, with only 0.55% in SU and 1.20% in multi-unit clusters. This indicates that a high portion of artifacts exhibit a biphasic shape (e.g., a sine-wave-like shape).

Moreover, our algorithm was able to detect more than half (50.77%) of all the events in clusters that were manually labeled as artifacts. The DER algorithm found a smaller but relevant proportion of duplicate spikes in clusters marked as single units (10.50%).

As a control, we analyzed the performance of our algorithm in the same dataset with clusters automatically labeled by Combinato and without manual evaluation. In this dataset, which consisted only of multi-units and artifacts, we detected 19.84% of all spike events (see also Figure A2). Most of the detected spike events in the automatically labeled clusters were also detected in the manually labeled dataset (95.91%). The small difference in the detected spike events affected mostly multi-units (80.32%) in the automatically labeled dataset. This difference was caused by the criterion to label spike events depending on unit classes (Table 2 and Table 3) since the number of clusters labeled as artifacts is lower in the automatically clustered data. This demonstrates that the DER algorithm is suitable for manually labeled as well as automatically sorted data. Therefore, the DER algorithm can be easily integrated in existing fully automated spike-sorting routines.

As the three parts of the algorithm can be run independently on the data, it is possible that spike events are detected in more than one part. To visualize the interactions, we created Venn diagrams of detection overlaps, separating the number of detected spike events into single, multi-units, and artifacts (see Figure 4D). As expected, the highest fraction of duplicate spike events was identified in clusters manually labeled as an artifact. While all three parts show an overlap in the detected duplicate spike events, there is an individual fraction of spike events that is detected within each part, underlining the importance of each step of the algorithm. Most duplicate spike events were detected based on the cross-correlations between clusters (Part III). This is the only step of the algorithm that uses the correlation of clusters across the entire time of recording and can therefore also identify a different population of duplicate spike events than the other parts that only compare simultaneous spike events. Note that Part I has a large overlap with the other two parts. This might be caused by artificial events (e.g., electrical noise) recorded simultaneously across several bundles which also fulfil the detection criteria of Part II or III.

### 3.3. Estimation of False-Positive Rate

In order to obtain an estimate for the false-positive rate of the DER algorithm, we used three different datasets (manually sorted original data, cluster-wise time-shifted surrogate data, and simulated data [58]). Figure 5 shows the percentage of detected spike events for these three datasets separately for different unit classes and for the different parts of the DER algorithm in which the spike events were detected. Percentages of detected events within the manually sorted data (original, blue bars) are the same as in Table 5 and are shown to visualize the comparison to actual false positives.

To estimate the false positive rate, we altered the manually sorted data by shifting each cluster circularly by a random offset time (cf. Figure 1B). The detected spike events are shown in red in Figure 5, indicating that the false positive rate is in a range of 0.01% with the notable exception of biphasic spike events in artifact clusters. This is due to the merging of artifact clusters in Combinato. Extracting spike events using both a positive and a negative threshold leads to biphasic spike events being extracted and sorted twice, but if both belong to an artifact cluster, they are merged into one cluster as there is only one artifact cluster per channel. Therefore, shifting the data in time by a constant random offset per cluster will not affect the detection of biphasic artifacts within each channel. All other comparisons between different clusters in the same channel are affected, which is reflected by an absolute decrease of 2.3% for artifacts detected by this part of the DER algorithm.

Finally, we estimated the false-positive rate using simulated data [58]. We used 80 individually simulated channels, each containing the activity of 2 to 20 neurons. Single units were computed using a Poisson distribution with a mean firing rate of 0.1–2 Hz (randomly selected), whereas the mean firing rate of multi-units was set to 5 Hz. No artifacts were included in this dataset (70.03% SU & 29.97% MU spike events). The mean firing rate of these simulated channels was 15.78 Hz. The number of detected spike events using our DER algorithm is shown in Figure 5 as yellow bars. The resulting percentages of false-positive detections are similar to those of the cluster-shifted data except for the biphasic events and the SU spike events detected by Part III. The simulation included only positive spike events, leading to zero detections of biphasic events in Part II of the DER algorithm. The higher fraction of spike events of single-unit clusters detected in Part III is due to the significantly higher firing rates in the simulated data compared to our original dataset (15.78 Hz vs. 4.64 Hz on average) and the larger number of SU clusters in the simulation. This does not affect the detection of multi-units in Part III because of the smaller fraction of multi-units in the simulated data. Thus, both the time-shifted surrogate data and the simulated data demonstrate that our algorithm produces a rather low percentage of false positive detection and thus operates at a rather high specificity.

As an additional control analysis, we repeated our DER algorithm on several spike extraction thresholds of Combinato (see Figure A3). Combinato’s standard extraction threshold of 5 σ seems to be optimal for our data, as the number of detected artifacts using the DER algorithm is strongly reduced compared to 4 σ, whereas only small changes appear when we increase the extraction threshold to 6 σ or 7 σ (while losing many low-amplitude spike events). 

The overall results of our algorithm (cf. Figure 4) demonstrate that human single-unit recordings contain a substantial percentage of duplicate spike events. The DER algorithm is an effective way to clean these events and increase overall data quality. 

## 4. Discussion

In this study, we have introduced a method to deal with the problem of coincident event detections in human single-unit recordings. We have demonstrated that coincident spike events appear considerably more often in actual recordings than expected based on a time-shifted surrogate distribution. In human single-unit studies, these duplicate spike events are typically not accounted for, although their removal represents a way to optimize data quality.

The clinical environment in which single units are recorded contains numerous sources of environmental noise. Different approaches can reduce the influence of noise, such as carefully optimizing every aspect of the setup, or identifying and eliminating sources of noise before the recording. It has been analyzed how different implantation-, cutting-, and splicing-techniques may improve data quality [23]. Nevertheless, in a clinical setup it is impossible to eliminate all disruptive influences. Therefore, identifying artificial influences in the data is essential. Algorithms used for the extracting, sorting, and clustering of spike events from neuronal activity recorded by microwire bundles incorporate basic artifact detection. Nevertheless, most of these algorithms do not take into account that artificial events often appear simultaneously on different recording channels. Simulated data are widely used to test spike-sorting algorithms because they allow a comparison of the resulting unit classification to ground truth [27,28,29,41]. However, this is not feasible for the development of our method as the different noise sources in a clinical recording setup cannot be convincingly simulated. Therefore, we employed a data-driven approach based on a large and reliable dataset of recordings that were manually reclustered. Due to the lack of ground truth telling us whether a detected duplicate spike event represents an artifact or a physiological phenomenon, we employed bootstrap approaches to arrive at reasonable assumptions.

The proposed DER algorithm primarily detected events in clusters that had been labeled as artifacts in our dataset. This observation underlines that our method is in good accordance with human operators. Nevertheless, manual clustering leads to a more conservative dataset, as an operator tends to label an entire cluster as an artifact if it is contaminated by many artificial events even though it might contain some neural events as well. We showed that manual reclustering of automated sorted data is not essential for the detection of duplicate spike events. This entails the possibility to run the DER algorithm also before manual reclustering, minimizing such contaminations and complementing manual evaluation. Alternatively, executing the DER algorithm after manual evaluation allows the algorithm to use the labeling information for detecting artificial or duplicated spikes (see also Table 2 and Table 3). Therefore, we recommend using the DER algorithm after a manual evaluation of clusters.

The evaluation of our algorithm on different surrogate datasets (cluster-wise time-shifted and simulated data) demonstrated a low false-positive rate and thus good specificity of the DER algorithm. The precise false-positive rate depends on many factors such as firing rate, number of channels and clusters, percentage of artifacts, etc. The best estimate for our setup is provided by the time-shifted dataset as it conserves these factors. The low percentages of resulting false-positive detections further encourages the use of our algorithm. 

Despite the convincing results of the presented method, we recommend that users take the DER algorithm with a grain of salt. For studies specifically investigating coincident spikes (e.g., [56,59,60,61]), the manipulations of the DER algorithm could be counter-productive. Nevertheless, it is important to note that our algorithm only deletes events within a rather narrow time window (Parts I, II, and III use 0.05 ms, 0.65 ms, and 0.5 ms, respectively; see Table 4). This still facilitates to measure certain asymmetries in the firing of neurons caused by direct synaptic connections [56] as well as common synaptic inputs [62]. The default parameters of the DER algorithm (see Table 4) can be easily adjusted to different recording setups and research questions. It is currently compatible to the spike-sorting algorithms Combinato spike sorter [29] and Wave_clus [27,41] and can easily be adjusted to others (for further information see https://github.com/Geaht/DER, accessed on 20 May 2021).

All data used in this study were recorded referentially against a low-impedance reference microwire that was stripped of insulation. Our spike-sorting program Combinato extracted spike events by independently applying both positive and negative detection thresholds. This led to biphasic spike events that are extracted twice if both amplitudes (positive and negative) exceeded the extraction threshold of our spike-sorting algorithm. In a recording setup that uses a bipolar montage, simultaneous biphasic events are likely to occur at a much higher frequency as a result of subtracting one channel’s activity from another. As the search for simultaneous biphasic spike events within a channel of our DER algorithm (Part II) is calibrated for referential recordings, we recommend adjusting this part of the algorithm for bipolar montages or skipping it completely. The findings of our study encourage future systematic investigation into how duplicate events are affected by wire bundle splicing and cutting. Furthermore, possible influences of different referencing techniques (e.g., local- vs. bipolar-referencing) should be further analyzed. In order to further optimize data quality, it is desirable to combine datasets from different recording sites and identify individual as well as shared noise sources. Including recordings from different amplifier types may also yield additional insights into the origins of artificial spike events. Today, the gold standard for single-unit recordings are intracellular recordings. Combining these with extracellular recordings (e.g., [26,46,47]) and focusing on artificial events that are recorded only on extracellular electrodes would allow us to further improve our understanding of noise sources.

We have demonstrated that, for recordings with microwire bundles in human patients, it is useful to examine possible interactions between different channels. Future single-unit studies should include similar algorithms to deal with this problem.

## Figures and Tables

**Figure 1 brainsci-11-00761-f001:**
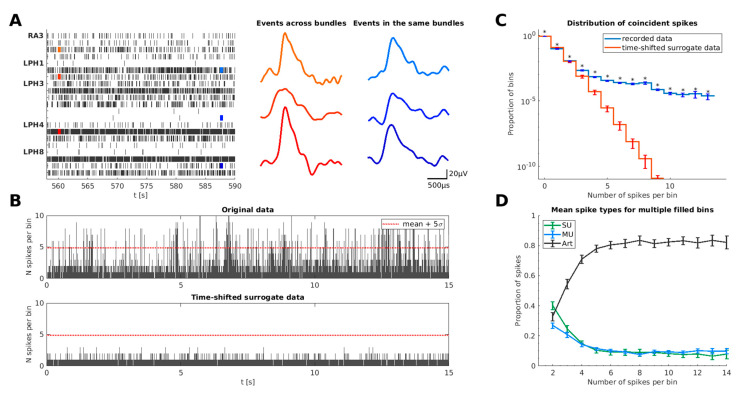
Characteristics of duplicate spike events in human microwire recordings. (**A**) Raster plot of units recorded from 5 microwires located in the right amygdala (RA) and the left posterior hippocampus (LPH). Red and blue lines indicate the timing of simultaneous spike-event shapes drawn next to the raster plot. The three red-shaded spike events occurred simultaneously across bundles while the three blue-shaded events occurred within the same bundle (left hippocampus); (**B**) Binned spike counts of a typical 15 s segment of data. Within each 0.5 ms time bin, the number of spike events across all 80 recording channels was counted. The red dotted horizontal line indicates the mean plus 5 standard deviations (σ) of all bin counts. A substantial fraction of time bins exceeds this 5σ-threshold, indicating a high proportion of simultaneous spike events. The lower panel displays a 15 s segment from the same recording session based on randomly circular time-shifted data for each cluster. In this time-shifted surrogate data, no bin count exceeded the 5σ-threshold of the original data. The *y*-axis is limited to 10 for better visualization; (**C**) Distribution of bins filled with different numbers of spike events across all 51 recording sessions. The blue curve shows the proportion of bins in the original recorded data, indicating a considerable number of bins with 5 and more spikes. The red curve shows the distribution for time-shifted surrogate data of all 51 recordings (1000 permutations, Wilcoxon signed-rank test, *N* = 51, *p* < 0.0036); (**D**) Distribution of spike-event types in filled bins. Spike events in bins with several spikes most frequently originated from artifact clusters. However, some of these events were also found in SU (single units) and MU (multi-unit) clusters (*x*-axis limited to 14 for which more than half of recordings contributed with at least 10 bins).

**Figure 2 brainsci-11-00761-f002:**
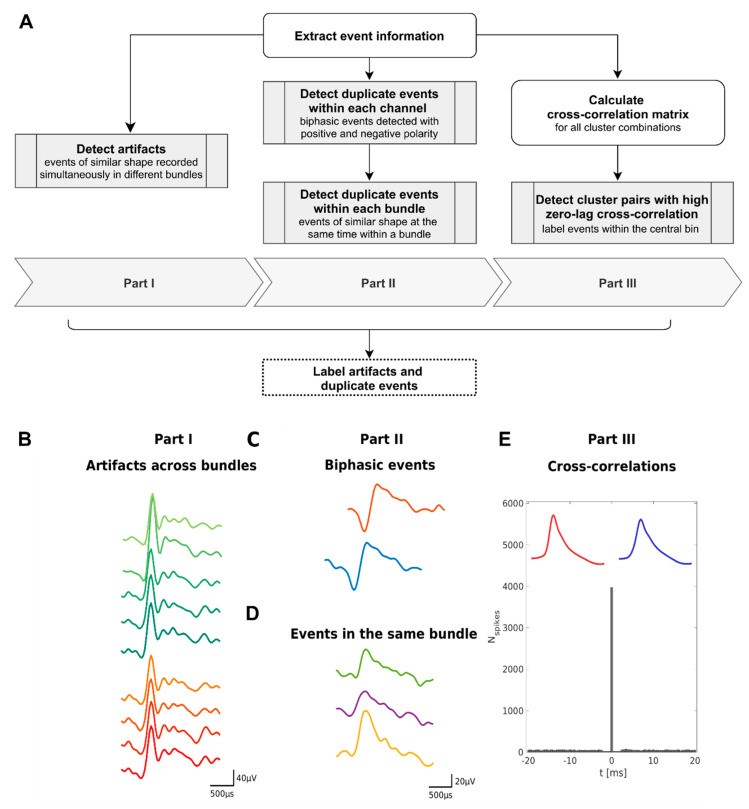
Schematic and illustrative examples for the three parts of the DER algorithm for the detection of spikes events that were recorded or detected multiple times. (**A**) Schematic flow chart of the algorithm. (**B**) In Part I we compare all combinations of the shapes of spike events appearing in a time window of 50 µs on different bundles. An example of event shapes recorded in different bundles is shown (red: left posterior hippocampus (LPH), green: left parahippocampal cortex (LPHC)). (**C**,**D**) In Part II, we first identify biphasic events that are detected with positive and negative polarity on the same channel within a time window of 650 µs. Panel (**C**) illustrates an example of biphasic event shapes (LPHC 5). Furthermore, similar events within the same wire bundle are identified if they appear within a time window of 50 µs and have highly similar shapes (example in (**D**) from LPHC). In Part III cross-correlograms are calculated for each combination of two clusters. If the central bin exceeds a certain threshold, the spike events within the central bin are considered to be duplicate. Subfigure (**E**) shows an example of the cross-correlograms of two single units that were recorded on two different microwires in the left amygdala. Both units have a large fraction of simultaneous spikes and a similar mean spike shape (shown in red and blue). For each part of our algorithm, we define criteria to determine which event to retain and which to label as artifacts (see also Table 2 and Table 3).

**Figure 3 brainsci-11-00761-f003:**
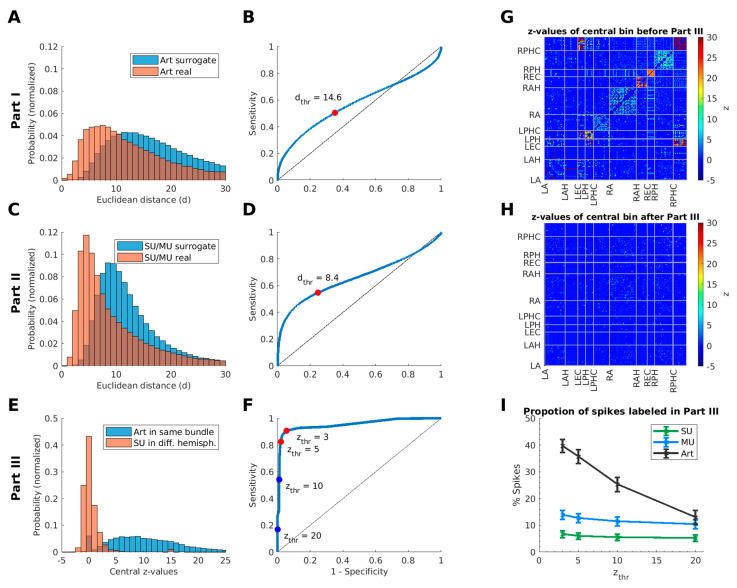
Selection of optimized parameters for the DER algorithm based on the recorded data. (**A**) Distributions of Euclidean distances between simultaneous artifact-cluster events (red) and between non-simultaneous artifact-cluster events randomly drawn from different hemispheres (blue). These distributions were used to determine the threshold of event shape similarity in Part I of the algorithm. (**B**) ROC curve for separating the two distributions of the Euclidean distances in Part I. The operating point on the ROC curve (closest point to the upper left corner) defines the threshold of the Euclidean distance (*d* = 14.6, marked by the red dot). (**C**) Equivalent distributions for joint single- and multi-unit clusters are plotted to define a threshold of the Euclidean distance used in Part II (detection within the same bundle). (**D**) The resulting ROC curve yields a threshold of *d* = 8.4 for Part II (marked by the red dot). (**E**) Distributions of z-values of the central bin in the cross-correlograms between single-unit clusters from different hemispheres (red) and artifact clusters within the same bundle (blue). The corresponding ROC curve is shown in (**F**), including operating points for different thresholds of the central z-value of the cross-correlograms. (**G**) Matrix of central z-values for all cross-correlograms from a recording session with 80 microwires (left (L) or right (R) hemisphere: amygdala (A), anterior hippocampus (AH), entorhinal cortex (EC), posterior hippocampus (PH), parahippocampal cortex (PHC)). Large z-values (red) indicate clusters with a large number of simultaneous spikes; (**H**) Matrix of central z-values for the same recording after removal of all spike events that were detected in Part III of the algorithm (*z*_thr_ = 5). Note that isolated z-values above 5 can result from changes in the background distributions of spike counts in the cross-correlograms. (**I**) Proportion of spikes detected in Part III of the algorithm for different cluster types (SU, MU, and artifacts), averaged across all 51 recording sessions, for different threshold values *z*_thr_. Error bars indicate standard error of the mean.

**Figure 4 brainsci-11-00761-f004:**
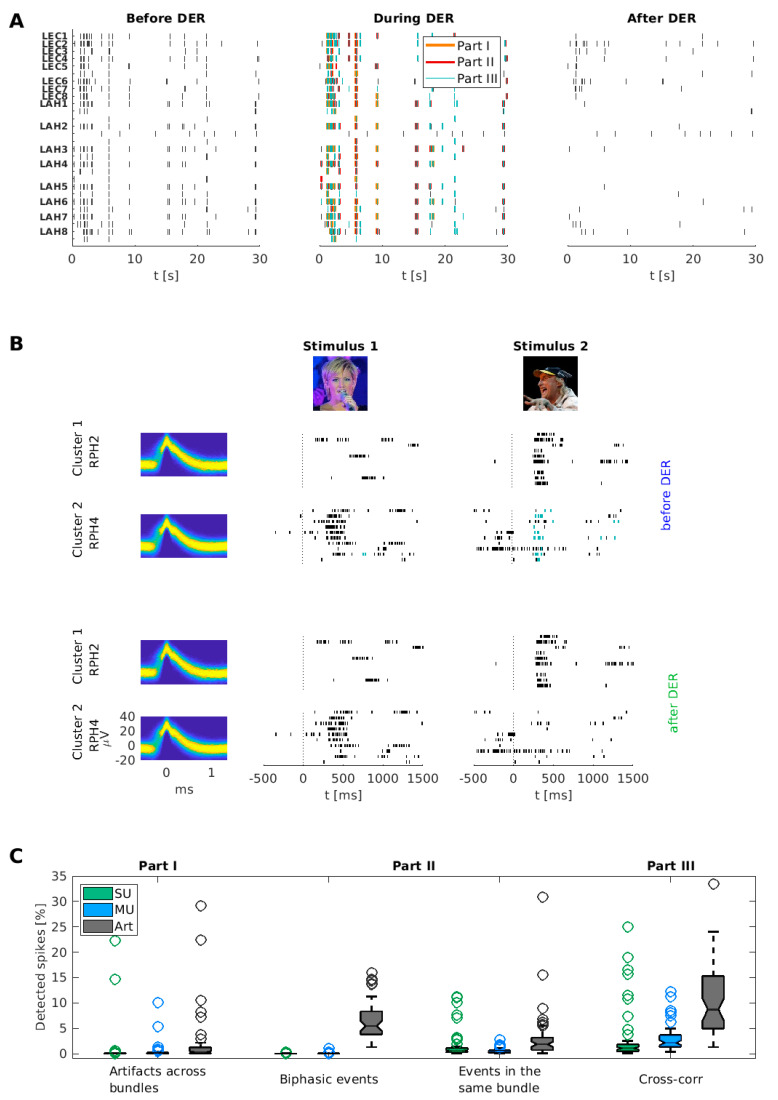
Performance of the DER algorithm applied to human single-unit recordings. (**A**) Exemplary raster plot of a recording before and after detection of artificial events using the DER algorithm. In the middle raster plot, all detected events are colored according to the part of the algorithm that detected them. (**B**) Raster plots of two units recorded from the same microwire bundle (right posterior hippocampus, RPH) that responded to two different stimuli (left: Stimulus 1—Helene Fischer; right: Stimulus 2—Otto Waalkes). The raster plots are plotted once with all extracted spike events (before DER—upper raster plots; spike events are marked according to the different parts of the DER algorithm using the same colors as in **A**) and once after removal of the duplicate spike events (after DER—lower raster plots). On the left, the spike waveforms of both clusters are shown as density plots. The first unit (RPH 2) responded to the image of the German comedian Otto Waalkes (Stimulus 2). This response displays no notable changes after applying the DER algorithm. The second unit on channel RPH 4 increased its firing rate to stimulus 1 and 2 in the original data. However, after deleting all events detected by the DER algorithm, the increase in firing rate to stimulus 2 was eliminated (red framed raster plot). (**C**) Boxplot of detected spike events (percentage of all spikes) separated by unit class and by the part of the DER algorithm. Part II is subdivided into the detection within the same channel and the same bundle. The *y*-axis is limited to 35% for visualization. (**D**) Percentage of detected spike events per unit class, separated into the three parts of the DER algorithm. Bold numbers indicate the percentage of spikes found in each individual part. Non-bold numbers in the intersections show the percentage of spike events detected by several parts. For the purpose of visualization, intersecting areas are not to scale.

**Figure 5 brainsci-11-00761-f005:**
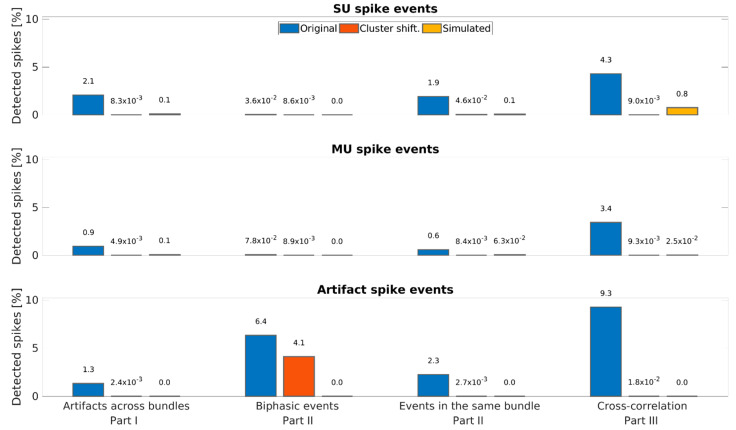
Estimation of the false-positive rate of the DER algorithm. Performance of the DER algorithm for three different datasets: original recorded data are shown in blue, cluster-wise time-shifted surrogate data are shown in red, and simulated data based on [58] are shown in yellow. The top panel shows the percentage of SU spike events found by the different parts of the DER algorithm. The middle and bottom panels show the same analyses for multi-unit and artifact clusters, respectively. The cluster shifted data as well as the simulated data are used to estimate the false-positive rate and thus specificity of the DER algorithm.

**Table 1 brainsci-11-00761-t001:** Overview of recorded sessions used to optimize the DER algorithm.

Patients	Recording Sessions	SU	MU	Art	#Events
13	51	2217	2212	4078	22,341,989

**Table 2 brainsci-11-00761-t002:** Criteria for labelling two coincident spikes events in Part II.

Case	Cluster Combination	Spike Events Labelled as Artifacts
1	pair of artifact and (SU or MU) within the same channel	spike event in the artifact cluster
2	pair of artifact and (SU or MU)	both coincident spike events
3	SU and MU in the same bundle	coincident spikes in the MU
4	two SU or two MU in the same bundle	coincident spikes in the lower SNR

**Table 3 brainsci-11-00761-t003:** Criteria for labelling coincident spikes events in Part III.

Case	Cluster Combination	Spike Events Labelled as Artifacts
1	pair of artifact and (SU or MU)	all coincident spike events
2	two clusters in different wire bundles	all coincident spikes
3	SU and MU in the same bundle	coincident spikes in the MU
4	two SU or two MU in the same bundle	coincident spikes in the lower SNR cluster

**Table 4 brainsci-11-00761-t004:** Default parameters of the DER algorithm.

Part	Parameter	Threshold	Value
I	min. number of simultaneous spike events	*no* _sim_	3
	max. time difference	∆*t*	50 µs
	max. Euclidean distance of spike-event shapes	*d* _thr_	14.6
II	max. time difference in the same channel	∆*t*_same channel_	650 µs
	max. time difference in the same bundle	∆*t*_same bundle_	50 µs
	max. Euclidean distance of spike-event shapes	*d* _thr_	8.4
III	width of time bins in the cross-correlograms	*t* _bin_	500 µs
	max. z-value of central bin count in cross-correlograms	*z* _thr_	5

**Table 5 brainsci-11-00761-t005:** Total percentage of spike events detected within each part of the DER algorithm (percentage of all extracted 22 341 989 spike events).

	Part I	Part II		Part III
Unit Class	Different Bundle	Same Channel	Same Bundle	Cross-Correlation
Artifacts	1.34%	6.35%	2.26%	9.27%
Multi-units	0.94%	0.08%	0.61%	3.45%
Single units	2.08%	0.04%	1.93%	4.30%

## Data Availability

The source code of the presented algorithm is publicly available on https://github.com/Geaht/DER, accessed on 20 May 2021.

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
