# Peer review of "Duplicate Detection of Spike Events: A Relevant Problem in Human Single-Unit Recordings"

_brainsci, 2021, doi:10.3390/brainsci11060761_

Round 1

Reviewer 1 Report

The manuscript ‘Duplicate Detection of Spike Events: A Relevant Problem in 2 Human Single-unit Recordings’ by Dehnen et al. describes an approach to a serious problem in multi-channel electrophysiology but fails to make a significant novel contribution towards a solution. This problem is what to do when a spike sorting pipeline detects two putative action potentials that occur at nearly or exactly at the same time. Typically, this results in a peak in the cross-correlation of the spike trains, which is readily detectable in all spike sorting curation software. The real question this paper is trying to address is how to determine if either or both of the simultaneously observed waveforms are indeed action potentials or not. While the manuscript describes this groups method for making this decision, the quantification of the performance is not valid because the ‘true positive’ labels in the ROC analysis are not as infallible as the authors suggest (see below: Specific Issues). Additionally, the results are not generalizable beyond their specific lab protocols and equipment. Therefore, overall the manuscript and does not constitute a publishable scientific finding. If the authors were to use proper ground truth and apply this method to diverse data sets and demonstrate utility in a broader setting, then I would consider taking another look.

Specific issues:

The manuscript relies too heavily on data and conclusions that are not quantified or presented in the paper. In particular, the manual labelling of spikes and artifacts as such by humans and the automatic detection of artifacts by Combinato is assumed to be infallible, leading to a conclusion that these labels are ‘ground truth’. This is not a sound conclusion, since there is no data presented to demonstrate that the labelling in fact perfect. Even if a perfect labelling is not available in human data, it is available in data from rodents and other animal models in which similar electrodes are used commonly and which contains experimentally evoked artifacts that do not require identification post-hoc.

Section 1.1.1 is a misrepresentation of the situation, in reality cross-correlation is sufficient for detection and is already included in most spike sorting pipelines including theirs.

Are the waveforms in Fig 1 spikes? Another problem is throughout the paper the authors go back and forth on whether the labels of spikes as such from Combinato are definitely spikes that may overlap with other spikes or rather maybe artifacts that overlap with each other or with real spikes. The SU/ME/ART labels are frankly not convincing enough to be used as they are in the paper as ground truth.

Fig 1 D is not meaningful because finding that the coincident spikes occur less after shuffling is trivial. The proper thing would be to compare not to a shuffled distribution but to a physiological distribution.

Author Response

Please see attached Word Document.

Reviewer 2 Report

The authors propose an algorithm to deal with multiple events detected in microwire recordings in humans. The topic is certainly relevant to the field and the results support the value of the proposed algorithm. I just have a few comments that I hope will improve the manuscript

1) The authors tested the algorithm with real data, which is essential for the topic under analysis. However, it would be interesting to see how the algorithm works when there are no artefacts (or very few of them). In a way, the question is about the false positive rate of the algorithm. Although it is not possible to have ground truth to assess this, perhaps you can run it again after removing every event originally labeled as artefact

2) Somehow linked to the previous comment, have you analised the source of the artefacts? For example, if they are due to line noise issues (a common problem), you could easily discard them by applying notch filters to the signal when you apply the filter for spike detection. In fact, it would be interesting to apply your algorithm to recordings with and without notches to see how many artefacts are removed on each case

3) When the authors say "cluster containing a high proportion of inter-spike intervals below the physiological refractory period most likely contains artifacts". This statement depends on the detection threshold used. If large, only real spikes from neurons should be detected, which then agrees with the authors idea that refractory violations will be from artefacts. However, if it is in a medium range, it is more likely that multiunit activity will be detected. Finally, using a low threshold will then also include the detection of random crossings of neural noise (from the activity of far away neurons), which I am not sure if they should be referred to as artefacts (in fact, there won't be coincidental activations across channels for these events so the proposed algorithm won't be able to deal with them). Moreover, the authors use a threshold of 5*sigma for spike detection and report that 25% of the events were labeled as artefacts. How is that number affected with a detection threshold of 4 or 7 times sigma?

4) The algorithm relies on an a priori labelling each detected event as MU, SU or artefact, and uses this as a ground truth. Although I do not believe that this can be done reliably, and even less so in an automated way, it would be good if the authors can comment on this and the effect on their proposed algorithm   

5) Fig 1 shows that coincidental events within and between channels are not really random. If you do the shuffling on a per channel basis o a per bundle basis (as opposed to on a cluster basis), you can quantify how much of the total number of coincidental events arise on each case, pointing to the relative a priori importance of the different parts of the algorithm.

6) The threshold of d=14.6 for section 2.3.1 is related to the distance between clusters in the feature space. How will the parameters of the filter applied before detection will affect this? For example if the low cut is 300Hz or 600Hz, spikes will have different energies and that will impact the distributions of wavelets coefficients. What about using a low detection threshold? Wouldn't the effect of more MU activity bring the clusters closer together for single units that are not of very large amplitude? Moreover, the detection threshold is applied to the median distance between all pairs of events within a time window that has events in at least 2 bundles. Is each pair composed by spikes from different bundles? What about some weight on the number of channels within a bundle that are showing a coincident spike event? If the artefact is spanning across different bundles it is also expected that most of the channels within the bundles will also exhibit that. Perhaps you could even analise a composite waveform by concatenating the event detected on each channel from a given bundle.

7) When discussing example 4B the authors emphasize the similarity of the waveforms recorded on different wires from the same bundle. If too different wires where actually close enough to listen to the same unit, the waveform could be very different. Still, Part III does not rely on waveform similarity so it should be able to overcome the double detection. In fact, which part was responsible for cleaning the raster on Fig 4B? I totally agree on the importance of removing these spurious responses as they affect any selectivity computation.

Minor comments
- At the beginning of page 7, there is a reference missing  
- Second paragraph of 2.5.2 references to section 2.4.2 and Table 2, but it seems those might be the wrong references. Same thing at the beginning of page 11.
- xlabel in Fig. A1 should be usec instead of msec

Author Response

Please see attached Word Document.
